# Telehealth Needs and Concerns of Stakeholders in Pediatric Palliative Home Care

**DOI:** 10.3390/children10081315

**Published:** 2023-07-30

**Authors:** Jannik Zimmermann, Marie Luise Heilmann, Manuel Fisch-Jessen, Holger Hauch, Sebastian Kruempelmann, Heidi Moeller, Laura Nagel, Michaela Nathrath, Vera Vaillant, Thomas Voelker, Merlin Jonas Deckers

**Affiliations:** 1Department of Psychology—Theory and Methodology of Counseling, University of Kassel, 34127 Kassel, Hesse, Germany; jannik.zimmermann@uni-kassel.de (J.Z.);; 2Palliative Care Team for Children Frankfurt, 60599 Frankfurt, Hesse, Germany; 3Palliative Care Team for Children Giessen, 35392 Giessen, Hesse, Germany; 4Palliative Care Team for Children Kassel, 34125 Kassel, Hesse, Germany

**Keywords:** telehealth, palliative care, home care services, pediatrics, needs, caregivers

## Abstract

Pediatric palliative home care (PPHC) provides care for children, adolescents, and young adults with life-limiting illnesses in their own homes. Home care often requires long travel times for the PPHC team, which is available to the families 24/7 during crises. The complementary use of telehealth may improve the quality of care. In this pilot study we identify the needs and concerns of patients, teams, and other stakeholders regarding the introduction of telehealth. As a first step, focus groups were conducted in three teams. For the second step, semi-structured interviews were conducted with patients and their families (*n* = 15). Both steps were accompanied by quantitative surveys (mixed methods approach). The qualitative data were analyzed using content analysis. A total of 11 needs were identified, which were prioritized differently. Highest priority was given to: data transmission, video consultation, access to patient records, symptom questionnaires, and communication support. The concerns identified were related to the assumption of deterioration of the status quo. Potential causes of deterioration were thought to be the negative impact on patient care, inappropriate user behavior, or a high level of technical requirements. As a conclusion, we define six recommendations for telehealth in PPHC.

## 1. Introduction

In Germany, children, adolescents, and young adults (hereafter referred to as “children”) with life-limiting or life-threatening illnesses are cared for at home by Pediatric Palliative Home Care (PPHC). PPHC provides 24/7 on-call palliative care at home. This improves the quality of life of children with life-limiting illnesses [1]. The highly specialized teams consist of physicians, nurses, social workers, and psychologists, and have large catchment areas by German standards; travel times of 90 min one way are not exceptional.

With PPHC, weekly home visits enable the team to assess the patient in the context of their complex illness, their care network, and their family system. Moreover, this helps build a relationship of trust, including the confidence the patients’ and caregivers’ feel towards the PPHC services [2,3]. Based on these palliative care assessments, teams are able to provide the necessary home-based support [3]. This means that, even in times of crisis, assistance can be rendered via phone without the need for a home visit in some instances [4]. Here we see the potential for improving the quality of care by means of the complementary use of telehealth, especially in providing assistance in crisis situations over physical distance [5,6,7,8,9].

Telehealth is an umbrella term for the exchange of medical information via electronic communication. It includes the aforementioned direct exchange between health care professionals, such as the multidisciplinary PPHC teams, and families and is also referred to as telemedicine or telenursing/telecare [7,10,11]. In addition, telehealth in pediatrics may include tele-education, tele-research, and disaster response [7].

Video consultation is already being used to provide care in general pediatrics and for critically and chronically ill children with dermatologic, cardiac, neurologic, psychiatric, and genetic conditions, such as those cared for in PPHC [7,8,9,12,13]. For example, video consultations and telemetry improved management and reduced interstage morbidity and mortality in infants with single-ventricle heart disease [8]. Synchronous or live communication can be used for video consultations as well as telemetry of medical measurements [7,8]. Peripheral devices such as patient monitors, stethoscopes, otoscopes, and ultrasound can be included for telemetry. Asynchronous or store-and-forward telehealth can also transmit telemetric data as well as messages including video, audio or images [7,8,14,15].

Initial reviews of pilot projects indicate that the use of telehealth in PPHC may be feasible and acceptable, especially in pediatrics, as most caregivers are younger adults with an increased ability regarding use of digital technology. However, a high-quality, controlled study is missing [5,6,16]. To further improve the quality of PPHC, the three teams in Hesse initiated the pilot project to develop, implement, and validate telehealth in PPHC in Hesse: “Pilotprojekt zur Konzeption, Implementierung und Validierung von Telemedizin in der spezialisierten ambulanten Palliativversorgung von Kindern, Jugendlichen und jungen Erwachsenen in Hessen” (TelPa_kids).

This article focuses on the needs assessment prior to the planned, complementary use of telehealth in PPHC. In a presentation, Lotstein et al. [17] proposed a stakeholder co-design-based survey of needs and requirements for telehealth. For the use of telehealth in PPHC in Los Angeles, an unpublished study identified three basic needs of patients, families, and PPHC team staff:information about PPHC for new patients;care network coordination;support services [17].

To our knowledge, there is no further research on needs and functional scope of telehealth in PPHC.

## 2. Materials and Methods

Following Lotstein et al. [17], we examined the perspectives of stakeholders in the German healthcare system, including patients, family members, teams, pediatricians in private practice, nursing homes, and ambulatory care nursing services. We used a cross-sectional design and a mixed methods approach, complementing qualitative with quantitative elements. We asked open questions to identify individual needs and concerns. Needs discussed in advance in the TelPa_kids research project were video consultation, telemetry, messenger services, data transmission, access to the patient file, symptom questionnaires, and information material. The interviewers explicitly asked about these needs in cases where they were not mentioned. A valence and intensity analysis were then conducted for the needs of the teams and families.

Information about the other members of the research team and the objectives of the focus groups and interviews were provided, and the results were made available to all participants after analysis. Authors L.N., M.L.H., H.M. and J.Z. are psychologists with experience in empirical social research, especially interviews and focus groups. Authors M.J.D., M.F.-J., H.H., M.N., S.K., T.V. and V.V. have proven expertise in pediatric palliative care and health services research. The authors M.J.D., M.L.H., L.N. and J.Z. received an interviewer training for this study to conduct interviews and focus groups. The study is registered in the German Clinical Trials Register under DRKS00030546. Approvals have been obtained from the Ethics Committees of the Universities of Kassel (EK-Nr. 202213) and Giessen (AZ 64/22).

### 2.1. Study I: PPHC Teams

#### 2.1.1. Participants

Focus groups were used as an economic method to assess the teams’ needs and concerns. In PPHC, the multidisciplinary team collectively establishes the everyday patient care decisions. Focus groups facilitate a dynamic data collection in which the relevance of a need or concern is discussed between the different professionals who know the actual conditions of their work. The individual opinions of specific members would not have been as able to capture the relevant needs and concerns. Accordingly, for the needs assessment of the teams, three members of the author team (M.J.D., M.L.H., J.Z.) conducted one focus group each for all of the teams in Hesse. Participation in the three focus groups was voluntary and took place on site (focus group K, focus group G) and via video conferencing (focus group F) during regular working hours. All staff on duty participated. The participants were already familiar with the TelPa_kids project and author M.J.D.

#### 2.1.2. Materials and Procedure

We developed the focus group guide based on a collaborative brainstorming session with the entire research team. The principal investigator (M.L.H.) had no contact with the teams prior to the study. The guide included information about (1) the process, (2) the framework, and (3) specific guiding questions. Three people (M.J.D., M.L.H., J.Z.) moderated the three focus groups, each of which lasted approximately 90 min.

The first of the two guiding questions was: “What are typical problems in everyday work where digitalization could help?” The second was: “What concerns do you have about the application of digitalization?” (see Table A1) The focus groups were held between February and March 2022. In May, a member check of the preliminary qualitative results was conducted with an additional quantitative survey.

### 2.2. Study II: Families

#### 2.2.1. Participants

To assess the needs of patients and families, we conducted a brief online survey using multiple choice answers and free text followed by semi-structured interviews (question guide can be requested from the authors). Participants were selected to ensure diversity of sociodemographic background and underlying conditions (stratified sample, see Table A2 and Table A3). We also sought diversity in native language and distance of residence to the PPHC team base. Families in acute crisis situations were not recruited. Participation was voluntary and unpaid.

Outreach was conducted through the three teams and was limited to individuals who were at that time receiving care or had received PPHC in the past. We recruited until material saturation occurred and no more new relevant aspects were mentioned in the interviews. The interviewer (L.N.) was unknown to the participants. Information about the aims of the survey, the TelPa_kids project, and the interviewer was provided in advance.

#### 2.2.2. Materials and Procedure

Between March and June 2022, patients and relatives were interviewed by means of an online questionnaire. The questionnaire included 9 items on sociodemographic information, 8 items on the care situation, and 9 items on needs for a telehealth app. This was followed by online interviews via Zoom “on premises” until June 2022 [18]. The average interview length was 23 min (range: 10–43 min). The semi-structured interview guide was also designed by the author team. It was strongly based on the guide used in Study I and the preliminary results of Study I. After an initial review of the material (*n* = 10), additional interviews were conducted until no new needs or concerns were raised by the families.

### 2.3. Study I & Study II: Qualitative Analysis

Based on the audio recordings and field notes of the focus groups and interviews, a structuring content analysis [19,20] and a valence and intensity analysis [19] were conducted using MAXQDA. One co-author carried out the analysis of each study (Study 1 = M.L.H.; Study 2 = L.N.). They first formed inductive categories of needs from the material alone. This was followed by matching to create a common category system to promote comparability. To show how the respondents rated the need categories, L.N. and M.L.H. conducted a complementary intensity analysis. For this purpose, we defined a 4-point scale from very high to low priority in order to map an evaluation and prioritization of needs ([19]; Table A4).

For quality control, 10% of the material was re-coded by another member of the author team based on the code book and the intercoder reliability was determined. The Cohen’s kappa according to Brennan and Prediger [21] was *κ_n_* = 0.85–0.90 for the needs. Overall, these values indicate good-to-very-good intersubjective comprehensibility of the code book.

### 2.4. Study III: Other Stakeholders

To identify the needs of other stakeholders involved in the outpatient care of chronically ill children, the authors M.J.D., M.F., H.H., S.K. and V.V. conducted a survey. Cooperation partners of the three teams in Hesse were interviewed about their needs regarding the complementary use of telehealth. Results were reported to M.J.D. and analyzed.

### 2.5. Study I-III: Quantitative Analysis

Quantitative analysis of the patient and parent survey, focus group member checks, and stakeholder survey were performed using MS Excel and SPSS version 26 (Table A5 and Table A7).

## 3. Results

### 3.1. Recruited Stakeholders

Overall, the three focus groups conducted in Study I had between 7 and 11 participants (*n* = 28) and consisted of physicians and nurses, supplemented by social workers, psychologists, and a team secretary.

A total of 15 families participated in Study II. All patients were cared for by their respective families, 2 were female and 13 were male, and the average age was 7.75 years (SD = 9.63, range = 0.4–25). The other sociodemographic characteristics and diagnoses (Table A2 and Table A3) were comparable to a German PPHC cohort study [13]. Of the 15 semi-structured interviews, three could be conducted directly with patients aged 24 to 25 years. In the remaining 12 cases, parents were interviewed. Three of the families interviewed had lost their child. Seven participants of the interviews were female and 9 were male, and the average age was 31.6 years (range = 24–46).

For Study III, we surveyed neuropediatric outpatient clinics (*n* = 1), pediatricians in private practice (*n* = 4), ambulatory care nursing services (*n* = 3), nursing homes and hospices (*n* = 5), and providers of medical aids (*n* = 1).

As telehealth is thought to enhance communication between stakeholders in PPHC, our results for needs, basic needs and concerns are presented for all stakeholders together.

### 3.2. Needs

The most important needs for the two user groups, families and the PPHC teams, are presented below. As the results show (see Table 1), there is a high degree of agreement between the two stakeholder groups on the priority of their needs—only 22.2% of the needs differed by more than one rank on a 4-point scale (see Table A4). Table A6 shows the results of the intensity analysis used to prioritize the needs. Needs prioritized as ‘very high’ by families and/or teams are:data transmission;video consultation;access to the patient records.

Needs prioritized as ‘high’ by families and/or teams were:
4.symptom questionnaires;5.communication support;6.shared calendar;7.informational materials;8.electronic stethoscope.

These priorities were confirmed in the patient/family online survey and team member check but were contradictory regarding the need for a messenger. In one focus group, the messenger was seen as a replacement for previous communication channels rather than an addition and was rated negatively. In the member check, the concerns and ambiguities were resolved. Thus, from the perspective of the teams, the messenger is seen as helpful, but not appropriate in emergency situations. In the online survey (see Table A7), 86.7% of the families indicated that they would like to communicate with the team via messenger, while in the interviews, the need for a messenger was only a medium priority. Table 1 and Table 2 provide an overview of the needs, their definitions, priorities, and stakeholder excerpts.

Figure 1 shows concrete applications and barriers for telehealth in PPHC as reported by families and teams. Needs with a lower priority (shared calendar, messenger) are not shown. The need for communication support was unanimously prioritized as “high” and is intended to address the problem of language barriers. Pictograms and plain or multilingual output were discussed here, without the need being further specified or taken into account in Figure 1.

The online survey of families in Study II provided additional insights. For example, 66.7% of parents and patients indicated their willingness to use telehealth apps, 26.7% were already using them, and one person (6.7%) was not willing to do so. In addition, the data provided insight into the preferences of individual stakeholders. For example, the three young adult patients showed the highest willingness to communicate with teams via messenger. The three bereaved caregivers were the only stakeholders who unanimously agreed that educational materials on “recognizing the dying process” and “actions in case of death” should be made available (see Table A7).

In addition, families also mentioned other features such as an emergency button with a locating function (5 families) or a “Frequently Asked Questions” relating to their own child to facilitate the communication of important information (5 families). Contrary to the authors’ expectations, telemetric transmission of vital signs (e.g., oxygen saturation, blood pressure) was not desired by the teams. The integration of an electronic stethoscope was seen as critical regarding technical feasibility. As an anecdote, we can report that after the needs assessment we tested the use of a Bluetooth-based electronic stethoscope by family members during video consultations with each of the three teams. A total of 9 patients receiving PPHC were evaluated, and 18 lungs examined. Of these, 7 lungs showed vesicular breath sounds, 6 lungs showed fine to coarse crackles, 2 lungs showed increased intensity of breathing sounds and one lung showed a lack of breathing sounds in its basal region. Only in one patient did audio quality prevent an evaluation of the lungs. However, the greatest obstacles proved to be the time lag between video stream and an auscultation signal, which itself also takes some time to get used to.

The other stakeholders in the care of chronically ill children were open to networking via an app, but no additional needs were mentioned. A company providing rehabilitation and orthopedic aids for children, nursing homes, and a hospice showed great interest in enhancing communication with the PPHC team, especially using video consultation, the electronic stethoscope, and access to the patient record. This need was not expressed by the neuropediatric outpatient clinic or the pediatricians in private practice.

### 3.3. Basic Needs

In a more in-depth study, four basic needs for telehealth were identified through content analysis, which aggregate the above needs. These can be generalized to both families and PPHC teams: (1) support: simplification of organizational and bureaucratic tasks; (2) support: more security in symptom reporting by the family; (3) overview and coordination of the care network; and (4) more effective and flexible communication.

### 3.4. Concerns

Barriers reported regarding implementation of telehealth in PPHC are shown in Figure 1. In addition, the families and teams also expressed general concerns regarding the introduction of telehealth (Figure 2). They are all based on the assumption of a deterioration of the status quo. Possible causes for a deterioration are seen in the negative impact on patient care, inappropriate user behavior, and a high level of technical requirements. The vast majority of concerns were expressed by the PPHC teams. Other stakeholders reported concerns regarding obtaining sufficient technical support.

There was also a concern that digitalization would replace traditional and ritual norms of patient care. One physician warned about the introduction of remote auscultation using electronic stethoscopes: “The auscultation with a stethoscope is a sacred act for patients. It makes a difference whether it is performed by a layman or a saint—the doctor is the best medicine!” Interestingly, families and teams did not express any concerns about data protection. Instead, they rated it as an obvious requirement for the app and expected that it would be sufficiently implemented.

## 4. Discussion

The use of telehealth in PPHC has been reported in pilot projects in hospices as well as in PPHC in Australia, where vast distances are a common factor [5]. These results are from the pre-SARS/COVID 19 pandemic era and do not include the patients’ perspectives. Due to the lack of a systematic needs assessment, no knowledge of stakeholder needs, and thus of the relevant functions for telehealth apps, is available [5,8,16]. Consequently, in the first part of the pilot project TelPa_kids, the needs assessment of all stakeholders was conducted as proposed by Lotstein et al. [17] via co-design and a mixed methods approach. Here, the basic needs reported by Lotstein et al. [17] could be extended and actualized for the German PPHC with 24/7 on call home visits within 2 h and were shown to be comparable to other surveys in pediatrics [7,22]. There was a wide range of 11 specific needs from ‘low’ to ‘very high’ priority. There was strong agreement between families and PPHC teams, thus we discuss these results together. The ranking of each priority differed in detail, which is understandable given the different perspectives of the stakeholders. For example, tele-education via the provision of information material has been shown to be helpful for patient and family education but needs careful curation and actualization by the teams [7].

From the most relevant needs for both user groups, only ‘video consultation’ and ‘symptom questionnaires’ have previously been reported as relevant for PPHC. Furthermore, in adult Palliative Home Care (adult PHC), the lack of a messenger function has been reported, whereas the high priority needs ‘data transfer’ and ‘access to patient records’ are new findings for PPHC [5,23]. The application of telehealth to cover these needs is sought after using functions tailored to the individual team. These should allow the integration of personalized questionnaires as well as team-specific restrictions regarding the access to patient records. This need also warrants the integration of telehealth into the electronic patient record of the team. The use of commercial telehealth providers in the context of general pediatric medical care can lead to fragmentation of health care because reports may not reach other care providers [7]. In the case of PPHC, there is a close exchange between co-providers and the team, yet content of telehealth care should definitely find its way into the team’s medical record to avoid fragmentation. Moreover, symptom questionnaires were rated as useful by families and teams for supporting symptom communication and control, as previously reported for pediatric and adult PHC [23,24]. Real-time patient-reported outcomes should thus also be made available in the patient record of the team.

Among the other stakeholders in the outpatient care for palliative children, the non-physician collaborators were particularly open to telehealth, especially video consultation with the teams. Our teams also identified telehealth and especially video consultations as an important tool for (inter-)professional exchange as described for other settings [7,15]. Data protection was seen as an obvious crucial factor for the use of telehealth and represents the most fundamental and challenging requirement for implementation [23].

Concerns about the implementation of telehealth in PPHC and pediatrics lay, as already previously reported, with the providers which worry about the negative effects on patient care [5,7]. Our findings validate and complement initial research on this topic [7,23,25,26]. Key concerns are fear of dehumanization and the loss of quality of patient care, as previously reported for adult palliative care [23,25]. Families and teams propose that these issues be addressed with clear rules for families and teams regarding the use of telehealth for home-based support; for example, that contact in crisis is made via telephone and not telehealth. Technological concerns were reported less prominently than in adult PHC, probably owing to a younger user age [23]. A particularly profound and new concern that emerged in our study was the reduction in the financial and human resources allocated to PPHC. Likewise, the loss of traditional face-to-face patient care was an important concern that was reported previously for adult PHC [23]. The reduction in face-to-face contact was also the most relevant of the few concerns expressed by families.

Video consultation is one of the most important needs. It presents the possibility of fundamentally changing the current standard of care by replacing all or part of the home visits with video consultation [23]. Accordingly, the majority of the concerns raised are also directed at video consultation. With this in mind, we would like to take a closer look at the potential value of video consultation. Given the way PPHC works, there are two distinct uses for video consultations: (1) for palliative care assessment; or (2) for home-based support of patients and families. Initial research on adult PHC suggests that video consultation does not appear to be appropriate for conducting the initial palliative care assessment [27,28,29]. In the context of PPHC with its much longer periods of care, routine weekly home visits for home-based reassessment take place in Hesse and many other German states. These face-to-face contacts enable the teams to give the required support to patients and caregivers. The identified concerns regarding loss of face-to-face time show that the limitation regarding the replacement of (re-)assessments through video consultations is also applicable to PPHC. By contrast, no concerns were raised in our study that video consultations would not be helpful in the home-based support of patients and caregivers, in addition to home visits and telephone contacts.

These results provide concrete information for the addressing of telehealth needs in PPHC and will lead to the technical implementation of an app in the next project phase. In addition, we plan to add the ability to collect data for clinical trials to the feature set of the telehealth app. Furthermore, our data indicate that future studies on telehealth in PPHC should assess the familial and teams’ sense of security of care during PPHC as well as familial empowerment as already discussed for adult PHC [23]. However, which specific app functions will find their way into the standard of care will be determined by each individual team, who will define the framework together with the families. The use of telehealth was also seen as promising by both families and teams during a break in PPHC or for follow-up care.

## 5. Conclusions

Considering the systematic stakeholder needs assessment, we have derived six recommendations for the complementary use of telehealth in PPHC. First, key functions should include data transmission, 3-party video consultations, access to patient records, symptom questionnaires, and communication support. Second, telehealth should be integrated into the team’s medical record. Third, telehealth apps should provide the ability for customized, personalized functions. Fourth, the impact of telehealth warrants review and adaptation of current practices as well as clear rules for families and teams regarding its use for home-based support. Fifth, palliative care assessments should be conducted as face-to-face contacts. Sixth, teams should network with other teams to ensure up-to-date and relevant information material and to conduct future controlled telehealth studies.

## Figures and Tables

**Figure 1 children-10-01315-f001:**
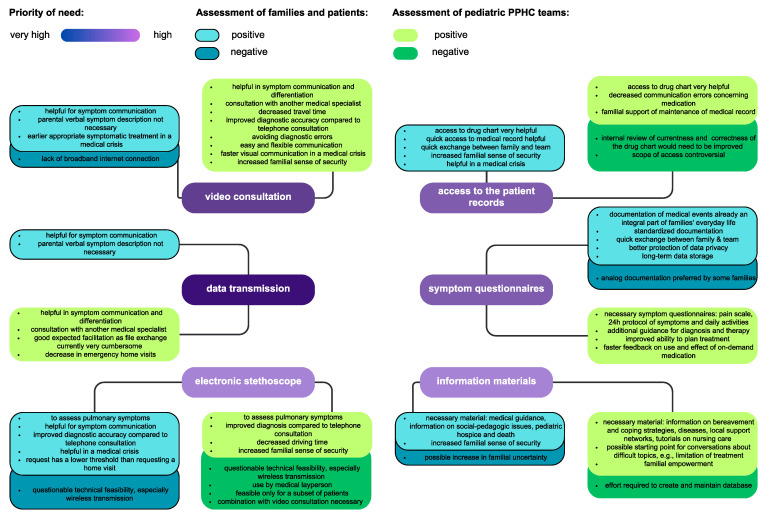
Illustration of applications and barriers to telehealth in PPHC.

**Figure 2 children-10-01315-f002:**
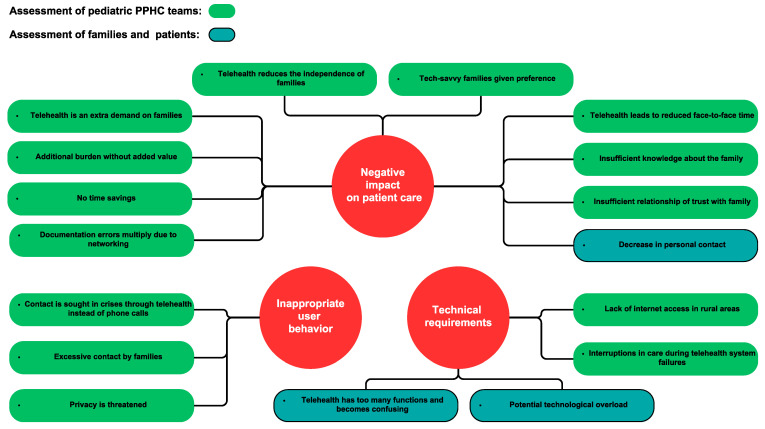
Concerns expressed by families and teams regarding the complementary use of telehealth in PPHC.

**Table 1 children-10-01315-t001:** Definitions and priorities for identified needs.

Need	Definition	Priority	
		PPHC ^1^ Teams	Families
Data transmission	Uniform, data protection-compliant transfer of files such as audio data, photos, videos or documents	Very high	Very high
Video consultation	Sound and image transmission in real time with at least 3 parties	Very high	High
Access to patient records	Families and other health care professionals can view and, if necessary, edit parts of the electronic PPHC patient records	Very high	Medium
Symptom questionnaires	Families document clinical progress (e.g., vital signs) or symptom scores, using online questionnaires to share with PPHC teams	High	High
Communication support	Picture and text material to overcome language barriers	High	High
Shared calendar	Shared online calendar to coordinate appointments between families and PPHC teams	High	Low
Information materials	PPHC teams provide families with current, relevant, and validated informational materials	Medium	High
Electronic stethoscope	Electronic stethoscope transmits auscultation audio signal in real time from family to PPHC teams	Medium	High
Messenger	Sending instant messages	Low ^2^	Medium

^1^ Pediatric palliative home care. ^2^ All focus group results (PPHC teams) were validated by a member check. For “Messenger”, the member check led to a group discussion which resolved concerns and resulted in a higher prioritization of a messenger.

**Table 2 children-10-01315-t002:** Excerpts for identified needs.

Need	Excerpt
Data transmission	Father: “In some cases you can’t explain well because of the language and [it] would be good to use the app to send videos and so on so that the doctors there can see and look at the case themselves.” (Family 15, para. 23 ^1^)
Video consultation	Nurse: “And especially when it’s the middle of the night, I would like to have a look first and you have a picture [with video consultation], you’re [then] right there, you can watch along and you don’t have to imagine it.” (FG 3, para. 68 ^1^)
Access to patient records	Mother: “Yes, maybe the possibility that you can print out a current medication schedule at short notice or look at it again, that would be great. Maybe also current doctor’s reports, because […] you always have to request it first, phone back and forth and send it back and forth. Something like that would be super practical.” (Family 2, para. 21 ^1^)
Symptom questionnaires	Mother: “That’s also quite good, because we were asked to observe and document. (…) So I have everything as a book (…) where I write down, for example, his seizures or abnormalities. So something like that is even better.” (Family 8, para. 41 ^1^)
Communication support	Doctor: “There are actually parents or nursing staff with whom you can’t speak a word of German. (…) But if it really were a reliable app and we could translate that one-to-one, that would of course be awesome.” (FG 3, para. 175, P5 ^1^)
Shared calendar	Nurse: “Hm, I would wish for: an app that you could use to coordinate the home visit appointments, because we plan a lot about how we should organize the home visits […] and then you make a phone call and find out the physiotherapy appointment is at that time […] and you have to change everything again.” (FG 1, para. 123 ^1^)
Information materials	Bereaved mother: “In particular, coping with grief or getting further support, for example, our stay at the rehabilitation center […] which we had to organize ourselves, no one gave us advice. Those would also be helpful points. And also funerals. You don’t know what to do. There could be a small guideline with tick boxes for what to do.” (Family 1, para. 49 ^1^)
Electronic stethoscope	Doctor: “Well, I think auscultation is more than the quality of the sound you hear. It’s also everything else you observe: How does he breathe in and out? How does he move? How motivated is he to participate? You would need very good visuals to do that. So I’m not sure if that’s helpful or if I’m actually more likely to initiate a home visit, because auscultation transmission makes me feel more uncertain.” (FG 1, para. 371 ^1^)
Messenger	Patient: “We’re not the kind of people who call the doctor about everything and say: Hey, I have this and that, but maybe it would simply be possible to chat with him and say: I’m not doing so well right now […] So if it’s not an emergency, then the doctor can check the message when he has time.” (Family 14, para. 29–31 ^1^)

^1^ Excerpts translated from German by the authors.

## Data Availability

The data presented in this study are available on request from the corresponding author. The data are not publicly available due to privacy issues. For further information on the TelPa_kids study see: https://www.kleine-riesen-nordhessen.de/kinderpalliativteam/telemedizin (accessed on 20 July 2023).

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
