# Peer review of "Telehealth Needs and Concerns of Stakeholders in Pediatric Palliative Home Care"

_children, 2023, doi:10.3390/children10081315_

Round 1
Reviewer 1 Report
I thank the Editor for the opportunity to review this manuscript.
The authors have addressed a relevant but still under-explored topic.
The description of the methods and results is accurate and comprehensive. In my opinion, this work is very praiseworthy for two reasons: the first because the sequence of the studies' phases is clear even to the reader less experienced in the field; the second because the Authors have tried to give a pragmatic slant to the results.
Only minor revision:
MATERIALS AND METHODS
- Line 111-114: the paragraph describes some results. Would it be possible for the Authors to include this data among the results?
- Line 123: supplementary Table B1. It is the first table of the appendix materials mentioned in the manuscript. To respect the appearing order, would it be possible for the Authors to reverse the order of all the tables of the supplementary materials? Make this table Appendix A, table A1, and the others Appendix B (B1-B5).
- Line 136-137: I think the sentence 'We recruited until material saturation occurred' needs more explanation.
- Line 137-140 and 142-144: the paragraphs describe some results. Would it be possible for the Authors to include this data among the results? In addition, I suggest to the Author to add the reference to supplementary materials tables (the current A1, A2 tables).
- Line 150-153: the paragraph describes some results. Would it be possible for the Authors to include this data among the results?
- It is unclear why the authors specify when and if the participants have familiarity/contact with the three members of the author team (M.J.D., M.L.H., J.Z.). Could you please add some clarification on this?
CONCLUSIONS
Conclusions in the current format seem a little too concise.
Reviewer 2 Report
The article needs minor amendments. I set out all my comments in the paragraphs below. Please refer to all of them, implement the amendments if necessary.
1. The introduction needs to be expanded slightly to include other aspects of telemedicine. Please go beyond the field of palliative medicine and write a few sentences based on the following quotations to be quoted in the text. This will enrich the value of the article.
10.3390/healthcare10102040
10.1542/peds.2015-1517
10.1097/MOP.0000000000001159
10.1177/0883073818807516
10.1089/tmj.2017.0041
2. The methodology of this study is almost well designed.... All the required elements including description of participants, procedure, Qualitative Analysis were applied. The statistical analysis of the results was forgotten? Why were no statistical calculations performed in this article to enrich the results and present them in a more meaningful and transparent way?
3. I am not convinced that the conclusions presented in the form of points are a good solution. Please convert this list into text. Alternatively, present the consensus of the study in the form of key points.
Round 2
Reviewer 2 Report
The authors have addressed all the guidelines from the previous review. Major changes have been made to the manuscript to allow acceptance in its current form. Thank you for your time.